# Identification and Validation of Quantitative Trait Loci Associated with Fruit Puffiness in a Processing Tomato Population

**DOI:** 10.3390/plants13111454

**Published:** 2024-05-23

**Authors:** Françoise Dalprá Dariva, Su Subode, Jihuen Cho, Carlos Nick, David Francis

**Affiliations:** 1Department of Horticulture and Crop Science, The Ohio State University, 1680 Madison Ave, Wooster, OH 44691, USA; fran_dariva@hotmail.com (F.D.D.);; 2Departamento de Agronomia, Programa de Pós-graduação em Fitotecnia, Universidade Federal de Viçosa, Av. P.H. Rolfs, s/n, Campus Universitário, Viçosa 36570-900, MG, Brazil; carlos.nick@ufv.br

**Keywords:** puffy fruit, marker-assisted selection, genomic selection, tomato breeding

## Abstract

Physiological disorders impact the yield and quality of marketable fruit in tomato. Puffy fruit caused by cavities inside the locule can be problematic for processing and fresh market quality. In this paper, we used a recombinant inbred line (RIL) and three derived processing tomato populations to map and validate quantitative trait loci (QTLs) for fruit puffiness across environments. Binary interval mapping was used for mapping the incidence of fruit puffiness, and non-parametric interval mapping and parametric composite interval mapping were used for mapping severity. Marker–trait regressions were carried out to validate putative QTLs in subsequent crosses. QTLs were detected on chromosome (Chr) 1, 2, and 4. Only the QTL on Chr 1 was validated in progeny from subsequent crosses. This QTL explained up to 22.5% of the variance in the percentage of puffy fruit, with a significant interaction between loci on Chr 2 and 4, increasing the percentage of puffy fruit by an additional 15%. The allele responsible for puffy fruit on Chr 1 was inherited from parent FG02-188 and was dominant towards increased incidence and severity. Marker-assisted selection (MAS) for the QTL on Chr 1 was as efficient as genomic selection (GS) in reducing the incidence and severity of puffy fruit, despite the potential contribution of other loci.

## 1. Introduction

The tomato is a major vegetable crop cultivated around the globe. Physiological disorders impact the yield and quality of marketable fruit and are of growing concern due to the increased severity of symptoms due to heat, drought, and excess moisture [1]. Physiological disorders are defined as defects, usually of fruit, and are exacerbated by exposure to abiotic stress conditions, including nutrient imbalances, temperature fluctuations, and water extremes. Cat-facing, blossom-end rot, yellow shoulder, cracking, and puffiness are examples of the physiological disorders of tomato. Genetic predisposition, plant nutrition, and production practices, including pruning and training, contribute to the occurrence of physiological disorders in tomato fields [2].

Puffiness is defined as the presence of open cavities between outer walls and locular contents in fruit that would normally be filled with gel, seeds, or placental tissue [3]. Because puffy fruit tend to be soft, ribbed, spongy, and gel-free, they are disliked by consumers of fresh tomato [4]. Grading standards for fresh market tomatoes set limits for puffiness [5]. Puffy fruit are also considered undesirable for the processing industry [6], with anecdotes of factory yield loss due to fruit floating during water aided sorting and exploding during heating in the peeling process.

Puffiness occurs due to either an overgrowth of the pericarp or poor development of the locule tissue [7]. Hence, any factor that impairs pollination, fertilization, or seed set promotes fruit puffiness [8]. Environmental conditions associated with fruit puffiness include high and low temperatures, excessive rain, low light intensity, and high nitrogen combined with low potassium levels [7,8,9]. Maintaining ideal temperatures for plant growth can minimize the impact of this disorder [4], which can only be achieved in indoor tomato production. Alternatively, breeding for tomato varieties that are less prone to puffiness may provide a solution for field-grown tomato. A genetic component is associated with fruit puffiness in tomatoes [1] and defining the genetic predisposition or resistance could provide a means to minimize this disorder in diverse tomato production systems.

Our specific goals were to describe the genetics underlying an important fruit quality trait and compare selection strategies. Previous efforts to map QTLs in tomato for physiological disorders such as cracking [10] and susceptibility to blossom-end rot [11] have been successful. Such genetic analysis also provides information relative to the location of loci, gene action, and the effect of an allele substitution, all of which may facilitate selection strategies [12]. QTL mapping is also an important step in prospecting or mining candidate genes [13]. Here, we describe a multi-generational analysis of the genetics underlying fruit puffiness within a processing tomato population. Three different statistical approaches were used to map QTLs for both the incidence and severity of fruit puffiness across two environments. QTL interactions increasing fruit puffiness severity were detected. Further, we compared the efficiency of strategies for selection based on marker-assisted selection (MAS) and genomic selection (GS) under different intensities, with both strategies showing promise to minimize puffiness in breeding populations.

## 2. Materials and Methods

### 2.1. Mapping Populations

The initial population used to quantify the occurrence of fruit puffiness and map quantitative trait loci (QTLs) was derived from a cross between Ohio 2K9-5533-1 and FG02-188. Ohio 2K9-5533-1 is an inbred line with a 50% fresh market Roma genetic background and a 50% processing tomato background. Ohio 2K9-5533-1 possesses resistance to tomato-spotted wilt virus (*Sw5*) and *Phytophthora infestans* (*Ph3*) [14]. FG02-188 is a promising tomato parent for commercial hybrids, with resistance to bacterial canker (*Clavibacter michiganensis* subsp. *michiganensis*). The population consisted of 159 F_5_ recombinant inbred lines (RILs).

Three populations derived from crosses to two commercial lines (UGP02 and UGP01), a bacterial spot resistant breeding line (OH813A), and three selections from the RIL population (2K19-8107-1, 2K19-8052-3, and 2K19-8022) were developed for QTL validation. The parent OH813A was previously described [15]. The hybrids, 2K20-8312 (2K19-8107-1 X UGP02), 2K20-8322 (2K19-8052-3 X UGP01), and 2K20-8357 (OH813A X 2K19-8022) were self-pollinated, and F_2_ progeny were advanced to produce 161 F_3_ generation families for evaluation and validation (53 families from 2K20-8312 and 2K20-8322 each, and 55 families from 2K20-8357).

### 2.2. Field Trials

Tomato seedlings were grown inside a glass-covered greenhouse with environmental conditions set to 24–26 °C daytime and 21–24 °C nighttime temperatures, a 12 h photoperiod, and a minimum light intensity of 190 μmol m^−2^ s^−1^ in the photosynthetic active range maintained using supplemental high pressure sodium lighting. Seeds were sown in 288-cell flats filled with a commercial growing medium (Promix FLX, Premier Horticulture Inc., Québec, QC, Canada). Water was applied daily as needed. Fertilizer applications started when seedlings were 2 weeks old and were performed once a week by adding a 220 ppm solution of Jack’s professional 20-20-20 (Scotts Sierra Horticultural Products, Co., Marysville, OH, USA) to the irrigation water. Field transplanting was carried out when seedlings were 4–6 weeks old.

Field trials to assess fruit puffiness were established in Wooster, OH, USA, where 154 tomato RILs were assessed, and Fremont, OH, USA, where 158 RILs were assessed, during the 2019 summer season. Plots were arranged according to an augmented experimental design. Three commercial tomato hybrids (PS696, UG16112, and H3402) were included as over-replicated checks in the trials to estimate experimental error. These checks were repeated eighteen times each throughout the tomato field to account for spatial variation across rows, columns, and quadrants. Each tomato plot was 6 m long and contained 20 plants spaced 30 cm apart. Rows were spaced 1.5 m apart. Plant care followed standard production practices used in the Midwest United States.

### 2.3. Fruit Puffiness Measurements

We measured fruit puffiness in nine randomly harvested fruit per plot. Fruit were cut longitudinally, scanned according to protocols outlined previously [16,17], and the images used to manually score fruit puffiness were both percentage ([number of puffy fruit/9]*100) and incidence of puffy fruit, with an image scored as 1, indicating the presence of puffy fruit, or 0, indicating absence.

### 2.4. Phenotypic Data Analysis

To investigate spatial variation in the field, the effects of genotype, rows, columns, and quadrants on the response variable percentage of puffy fruit, we fit a linear regression model via maximum likelihood estimators using the lmer() function in the lme4 package in R [18]. The full experimental model was as follows: y=u+GEN+ROW+COL+QUAD+e, where y was the vector of phenotypic observations, e.g., the percentage of puffy fruit; u was the model’s overall mean; GEN was the effect of SNP genotype; ROW was the effect accounting for variation across rows consisting of a single tier of plots; COL was the effect accounting for variation across columns consisting of three beds; QUAD was the effect accounting for variation across quadrants; and e was the residual error, with all these effects considered random. The variables ROW, COL, and QUAD describe a three-dimensional grid with over-replicated checks used to estimate variation. We used the ANOVA (model 1, model 2) function and syntax to compare the full model (model 1) against a second model (model 2), with either GEN, COL, ROW, or QUAD dropped to test the significance of the effect in question. *p*-values of the chi-square statistics equal to or lower than 0.05, obtained from maximum likelihood ratio test comparisons, indicated whether the dropped effect was significant. Additionally, correlation analyses were used to investigate the relationship between percentage and incidence of puffy fruit across the Wooster and Fremont environments.

### 2.5. DNA Isolation and Genotyping

DNA was extracted from young leaves using the modified cetyltrimethylammonium bromide (CTAB) method described in [19]. DNA samples were then genotyped using a 384 SNP panel with loci optimized for use in cultivated tomato based on recombination, genome coverage, and polymorphism rates [20,21,22]. The amplicon-based genotyping by sequencing PlexSeq^TM^ platform was used for SNP genotyping (AgriPlex Genomics, Cleveland, OH, USA). SNP calls were phased as “A” and “B” in the RIL population, and “A”, “B”, and “H”, in the validation population, in which “A” and “B” designate homozygotes for the FG02-188 and Ohio 2K9-5533-1 alleles, respectively. Heterozygotes were designated “H” in the validation population and set to missing data (NA) in the RIL.

### 2.6. Genetic Linkage Map Construction

Data quality control consisted of removing monomorphic markers, markers and individuals with more than 50% missing genotypic data, and markers with segregation distortion detected at *p* < 0.0001. We removed 15 markers with failed SNP calls for the reference parent FG02-188, 198 markers that were monomorphic in the RIL population, 3 markers with failed SNP calls for more than 50% of the progeny, and 32 markers with strong segregation distortion (*p* < 0.0001). No individuals were removed.

A genetic map was formed using the R/qtl package [23] in R statistical software, version 4.1.2 [24]. The 136 SNP markers were grouped using R/qtl and compared to their physical position in the Tomato Genome version SL4.0 [25]. The function est.rf()was used to estimate recombination frequencies between markers. Marker order was estimated through the orderMarkers() function with the map.function = “kosambi” argument. The Kosambi mapping procedure converts recombination frequencies into genetic distances (cM) and has the advantage of correcting for multiple crossovers [26]. Chromosomes were split into two linkage groups if recombination fractions between markers within a chromosome were greater than 0.5 and the LOD scores for the test of recombination fractions were less than 2. The ripple() (window = 4 and method = “likelihood”) and switch.order() functions were used to compare the initial marker order with alternative orders, including the marker order in the reference genome (Tomato Genome version SL4.0), based on logarithm of odds (LOD) scores and chromosome length. Higher LOD scores and lower chromosome lengths were observed when markers were arranged in the same order as in the reference genome and were therefore preferred. The quality of the final map was assessed based on genome coverage, linear regression, and by rank correlation analyses between marker position in the genetic linkage map and the physical position in the physical map version SL4.0 [25].

### 2.7. QTL Mapping

QTL mapping for fruit puffiness within the RIL population was performed for percentage and incidence of puffy fruit in the two locations (Wooster and Fremont) separately and combined, using three different statistical approaches. Genotype probabilities were estimated using the calc.genoprob() function in R/qtl with a step size of 1 cM. QTLs for the incidence of puffy fruit were detected through interval mapping using the scanone() function and the model = “binary” argument (IM-BI) in R/qtl [23]. For the percentage of puffy fruit, QTLs were detected through interval mapping using the scanone() function and a non-parametric model (model = “np” argument) (IM-NP) since residuals were not normally distributed according to the Shapiro–Wilk test (*p* < 0.05). Composite interval mapping (CIM) using the cim() function with one marker covariate and Haley–Knott regression [27] as the solution-generating algorithm was also used to detect QTLs for the percentages of puffy fruit. The non-parametric model and CIM were compared to verify CIM sensitivity to the violation of normality assumptions, especially because this method has recently been widely implemented, as seen in [28,29,30]. LOD significance thresholds to declare a QTL were determined by permutation tests (α = 0.05, n = 1000; [31]). We defined QTL location based on the 1-LOD support interval (LSI) from the LOD peak (a maximum LOD score in a chromosome that is above the LOD cutoff at the 0.05 significance level). The percentage of variance explained (PVE) and additive effects of QTLs for the percentages of puffy fruit were based on the nearest marker to the QTL peak position and estimated using the makeqtl() and fitqtl() functions of R/qtl [23]. PVE was given by the formula 1–10^−2 LOD/n^, where n is the sample size and LOD is the LOD score. Additive effects were equal to half the difference between the phenotype averages for the two homozygotes. A univariate binary logistic regression was fit for fruit puffiness incidence data using the glm() function and family = binomial argument in the R Core package. Odds ratios were expressed as $e^{\beta_{1}}$, where e is the natural base and equals to 2.718 and $\beta_{1}$ is the regression coefficient for the homozygote A.

### 2.8. Interaction between QTLs

Interactions between QTLs were investigated using a linear model ANOVA for all the nearest marker combinations using the “lm” function in the R core package [24]. The linear model used was $y=u+M_{x}+M_{y}+M_{x}:M_{y}+e$, where y is the vector of phenotypic observations, in this case, the percentage of puffy fruit; u is the model’s overall mean; $M_{x}$ is the effect of the nearest marker to the QTL peak position on chromosome x (with x being any chromosome with a QTL for the percentage of puffy fruit); $M_{y}$ is the effect of the nearest marker to the QTL peak position on chromosome y (with y≠x, and equal to any chromosome containing a QTL for percentage of puffy fruit); $M_{x}:M_{y}$ is the interaction effect between $M_{x}$ and $M_{y}$; and e is the random error, with all effects considered fixed except the random error. Tukey’s Honest Significant Difference test at 0.05 significance was used to compare the means for the percentage of puffy fruit between all genotype combinations, in this case, $A_{M_{x}}A_{M_{y}}$, $B_{M_{x}}A_{M_{y}}$ $A_{M_{x}}B_{M_{y}}$, and $B_{M_{x}}B_{M_{y}}$, in which $A_{M_{x}}$ and $A_{M_{y}}$ are homozygotes for the parent FG02-188 allele at marker $M_{x}$ and $M_{y}$, and $B_{M_{x}}$ and $B_{M_{y}}$ are homozygotes for the alternative allele at marker $M_{x}$ and $M_{y}$, respectively. Mean comparison tests were conducted using the HSD.test() function of the agricolae package in R [32].

### 2.9. QTL Validation and QTL Action

Validation was conducted using F_3_ families grown in Fremont during the 2022 summer season. Seedling growth, plot size, plant care, and fruit puffiness measurements followed the same procedure described for the RIL population. The measurement of fruit puffiness and incidence of puffiness and genotyping were also as described for the RIL. A total of 198 out of the 384 SNP markers were polymorphic for the validation families. We performed QTL validation by fitting linear regression models between the variable percentage of puffy fruit and each marker potentially linked to a QTL for fruit puffiness according to interval mapping results (or within the 1-LSI QTL interval) using the lm() and anova() functions of the R core package [24]. We considered cross as a covariate in the regression analysis so that the model was denoted as y=u+G+C+e, where y was the percentage of puffy fruit, G and C were the genotype and cross effects, respectively, and e was the random error. Sub-populations were also analyzed separately using the same model with the cross effect dropped. A QTL was considered validated when the effect of a marker previously linked to a QTL was significant by the linear ANOVA’s F-test at 0.05 significance level with a Bonferroni correction applied.

The presence of heterozygotes in the validation population allowed us to investigate QTL action through mean comparison tests between genotype classes. A Tukey’s Honest Significant Difference test at 0.05 probability level was conducted to compare the percentage of puffy fruit averages between homozygotes A, homozygotes B, and heterozygotes for validated markers using the HSD.test() function of the agricolae package in R [32].

### 2.10. Comparison of Selection Strategies for Decreased Fruit Puffiness

To investigate whether MAS for validated QTLs would result in decreased fruit puffiness, we compared the percentage of puffy fruit averages of randomly selected families from the validation population with that of families selected through MAS, GS, and the combination of GS and MAS (GS and MAS), under selection intensities of k = 1.66 (10%) and k = 0.80 (20%). The MAS strategy consisted of randomly selecting families that were homozygous for the allele B of solcap_snp_sl_20440 and solcap_snp_sl_18619, the two markers most likely to be linked to the QTL on chromosome 1. The GS strategy consisted of selecting families with the lowest genomic estimated breeding values (GEBVs) for percentage of puffy fruit. For the GS and MAS strategy, instead of randomly selecting families that were homozygous for the allele B of solcap_snp_sl_20440 and solcap_snp_sl_18619 markers, we selected those with the lowest GEBVs for the percentage of puffy fruit. k = 1.66 and k = 0.80, in this case, mean that 16 and 32 out of 162 families from the validation population were selected, respectively. The means of selected families were compared by the Tukey’s Honest Significant Difference test at a = 0.05 significance level [32].

The GS model used to predict GEBVs for the percentage of fruit in the validation population was developed using genotypic and phenotypic data from the mapping population, *Cmm*-RIL. Model development included genotypes for 163 SNPs and the percentage of puffy fruit averages for 149 *Cmm*-RIL lines across two locations. SNP calls were phased as ‘−1, 0, 1’, referring to homozygotes for the allele of Ohio 2K9-5533-1, heterozygotes, and homozygotes for the allele of FG02-188, respectively. Such a change is a requirement of the rrBLUP package used to run the GS model. A call rate > 80% and MAF > 0.05 were the criteria adopted for quality control of SNPs. The few NAs left were replaced by the most frequent mode. The effects of markers were estimated by the mixed.solve() function of rrBLUP [33] in R [24]. mixed.solve() fitted the ridge regression model y=Xβ+Zu+ε to the data, where y is the corrected phenotypic data for each line, X is the design matrix for the fixed effects β (which was not provided here), Z is the genotype matrix for the random marker effects u with $u\sim N\left(0,K\sigma_{u}^{2}\right)$, K is a positive semidefinite matrix (an identity matrix in this case), and ε is the error vector [33]. rrBLUP assumes that all markers have some effect on the trait and share a common variance [34]. Fruit puffiness GEBVs of validation families were estimated by simply multiplying the vector of marker effects with the genotype matrix. GS model quality and predictive ability was checked by leave-one-out cross-validation and linear correlation between predicted GEBVs and measured the percentage of puffy fruit averages for the two locations.

## 3. Results

### 3.1. Fruit Puffiness Assessment

The symptoms of fruit puffiness are shown in Figure 1. A total of 66% of the RILs (101 out of 154) had no puffy fruit in the Wooster field trial, while 59% lacked puffy fruit in the Fremont trial (93 out of 158). Sixty-six RILs had no puffy fruit in both locations. There was a significant correlation between locations (r = 0.43, *p* < 0.001), suggesting that RILs with a high percentage of puffy fruit in Wooster tend to have a high percentage of puffy fruit in Fremont and vice versa. Based on incidence, RILs with puffy fruit in Wooster also tended to have puffy fruits in Fremont (r = 0.21, *p* = 0.01).

Genetic causes were associated with variation in fruit puffiness within the RIL population based on a pairwise comparison of models (*p* = 0.0026). Row, column, and quadrant effects, on the other hand, were not significant (*p*-values ranging from 0.18 to 1.0), and therefore, no correction of the original data for spatial variation was needed for the QTL analysis.

Fruit puffiness was more severe in the validation populations than in the mapping population. Validation families had an average percentage puffy fruit of 26.7% compared to 7.1% and 10% observed for the RILs in the Wooster and Fremont field trials, respectively. A variation in the percentage of puffy fruit was also observed among the validation sub-populations. The average percentages of puffy fruit for 20K20-8312, 2K20-8322, and 2K20-8357 were 42.8%, 27.9%, and 10.1%, respectively.

### 3.2. Linkage Map Quality

The quality of the linkage map was assessed based on genome coverage and correlation with the physical map. Table 1 summarizes marker distribution across the 12 tomato chromosomes. Chromosome (Chr) 4 was split into two because the markers solcap_snp_sl_4042, solcap_snp_sl_47843, and solcap_snp_sl_4139 showed recombination fractions greater than 0.5 and test LOD scores lower than 2 relative to other Chr 4 SNPs. The number of markers per chromosome ranged from 3 on Chr 11 to 30 on Chr 4 (27 on 4a and 3 on 4b). The lowest and highest chromosome length estimates ranged from 11.1 cM for Chr 11 to 93.3 cM for Chr 6. The average distance between markers was 4.9 cM, and the largest distance between markers was 51.4 cM, observed on Chr 9. The marker physical position in the tomato Sl4.0 physical map [25] agreed with the estimated genetic position in the linkage map (Table 1). All rank correlation coefficients were equal to 1.00 (Table 1). The R^2^ for linear regressions was high for all linkage groups except 11 and 12 and ranged from 0.48 to 0.99 (Table 1). Poor fit of the data on groups 11 and 12 is likely due to the small number of markers for these linkage groups (3 on Chr 11, and 5 on Chr 12).

### 3.3. QTL Identification

QTLs associated with fruit puffiness were detected in Fremont and in the combined data, but not in Wooster. The number of QTLs observed among mapping approaches was consistent, with differences reflecting the environment in which data were collected and the confidence interval rather than the chromosome location, method of scoring the trial, or statistical approach used. For the percentage of puffy fruit, the non-parametric model (IM-NP) detected a QTL located between 11 and 38 cM on Chr 1, a QTL between 7 and 38 cM on Chr 2 in Fremont, a QTL between 14 and 38 cM on Chr 1, and another QTL between 22 and 37 cM on Chr 2 using data for both locations (Figure 2). CIM detected QTLs on Chr 1, 2, and 4 between 9 and 30 cM; 28 and 37 cM; and 0 and 5 cM, respectively, in Fremont, and on Chr 2 between 30 and 36 cM in both locations (Figure 2). As for the incidence of puffy fruit, the binary model (IM-BI) detected a QTL on Chr 1 between 11 and 38 cM in Fremont, and on Chr 1 and 2 between 16 and 39 cM, and 22 and 39, respectively, in both locations (Figure 2). The one-LOD support intervals (LSIs) estimated on Chr 1 and 2 using different mapping approaches overlap (Figure 2), which suggests that different mapping approaches are all detecting the same QTL. The SNP markers solcap_snp_sl_18619, and solcap_snp_sl_4963 on Chr 1, solcap_snp_sl_13625 and solcap_snp_sl_23850 on Chr 2, and solcap_snp_sl_21372 on Chr 4 appear within 1-LSI QTL intervals for all mapping approaches (Figure 2) and are therefore more likely to be linked to these respective QTLs. In most cases, B was the beneficial allele (Table 2), suggesting that FG02-188 was the parent responsible for increased fruit puffiness in the RIL population. The allele from the parent Ohio 2K9-5533-1 on chromosome 1 decreased the percentage of puffy fruit by 3.1 to 4.4%, and on Chr 2 by 4.1 to 5%. On Chr 4, the allele from the parent Ohio 2K9-5533-1 increased the percentage of puffy fruit by 4.7%. The QTL on Chr 1 explained between 5% and 6.4% of the total variation in the percentage of puffy fruit (Table 2). On Chr 2, the QTL explained from 8.3% to 9.5% of the variation in the percentage of puffy fruit (Table 2). The QTL on Chr 4 explained 7.1% of the variation in the percentage of puffy fruit in Fremont (Table 2). The odds of having puffy fruit for homozygotes A on Chr 1 are 3.4 to 3.6 times the odds for homozygotes B. The odds of having puffy fruit for homozygotes A on Chr 2 were 3.9 times the odds for homozygotes B.

### 3.4. QTL Validation and Interaction

Significant marker–trait associations detected in the validation populations confirm only QTL on Chr 1 (Table 3). Linear regressions for percentage of puffy fruit on the markers solcap_snp_sl_33701, solcap_snp_sl_8669, solcap_snp_sl_100164, solcap_snp_sl_20440, solcap_snp_sl_18619, solcap_snp_sl_4963, and solcap_snp_sl_531 located within 1-LSI on Chr 1 (Figure 2) were highly significant (*p* < 0.0001) in the validation population. Linear regressions for percentage of puffy fruit on markers within the 1-LSI QTL interval on Chr 2 (*p*-values ranging from 0.76 to 0.80) and 4 (*p* = 0.94), on the other hand, were not significant.

Because the cross effect was significant for all marker–trait regressions, we also analyzed marker–trait relationships for the three sub-populations separately. Potential QTLs on Chr 2 and 4 were also not supported (Table 3). Marker–trait associations for QTL on chromosome 1 were considered non-significant in at least one sub-population (Table 3). Since regressions upon solcap_snp_sl_20440 and solcap_snp_sl_18619 had the highest significance values (*p* = 3.02 × 10^−11^) in the combined validation population, they could be useful tools for marker-assisted selection. Interestingly, solcap_snp_sl_20440 and solcap_snp_sl_18619 were monomorphic for the beneficial allele B in the population 2K20-8357 with the lowest percentage of puffy fruit (10.1%) and monomorphic for the allele A in the population 2K20-8312 with the highest percentage of puffy fruit (42.76%). solcap_snp_sl_20440 and solcap_snp_sl_18619 explained 22.5% of the total variation in percentage of puffy fruit observed (R^2^ = 0.225).

An interaction between QTLs on Chr 2 and 4 may increase the percentage of puffy fruit. A significant interaction was observed between solcap_snp_sl_13625, the nearest marker to the QTL peak position on Chr 2, and solcap_snp_sl_21372, the nearest marker to the QTL peak position on Chr 4, for the Fremont data and combined data (*p* < 0.05). Although the interaction was not significant for the Wooster data (*p* > 0.10), genotype mean values were numerically consistent with an interaction. RILs with the marker combination, $A_{M_{13625}}B_{M_{21372}}$, homozygous for the allele A of solcap_snp_sl_13625, and homozygous for the allele B of solcap_snp_sl_21372 had a significantly higher percentage of puffy fruit in Fremont and in the data combined from both locations compared to other allele combinations (Figure 3). Although not statistically different for Wooster data, the percentage of puffy fruit in $A_{M_{13625}}B_{M_{21372}}$-RILs (12.3%) was on average higher than that of $A_{M_{13625}}A_{M_{21372}}$ (7.5%), $B_{M_{13625}}A_{M_{21372}}$ (2.8%), and $B_{M_{13625}}B_{M_{21372}}$ (4.4%) (Figure 3).

The validation population was not appropriate to confirm the existence of an interaction between Chr 2 and 4 because the genotype classes in it were not evenly represented. There were 24 $A_{M_{13625}}A_{M_{21372}}$ and 33 $A_{M_{13625}}B_{M_{21372}}$ families, while only 2 $B_{M_{13625}}A_{M_{21372}}$, and 5 $B_{M_{13625}}B_{M_{21372}}$ families were represented in the validation population.

### 3.5. QTL Action

The QTL on Chr 1 exhibits complete dominance for increased fruit puffiness. Figure 4 contains mean comparisons for the percentage of puffy fruit by genotype for the markers solcap_snp_sl_33701, solcap_snp_sl_8669, solcap_snp_sl_4963, and solcap_snp_sl_531, located on Chr 1. Linear regressions for the percentage of puffy fruit on these markers were significant in the field validation (Table 3). The average percentage of puffy fruit between homozygotes A and heterozygotes did not differ, but it was significantly lower for homozygotes B (Figure 4). No conclusions could be drawn based on the markers solcap_snp_sl_20440 and solcap_snp_sl_18619 because they were either homozygous A or homozygous B for all families.

### 3.6. Marker-Assisted and Genomic Selection

Based on our results, QTLs explained at most 22% of the variation for the percentage of puffiness, which suggests that a QTL below the detection limit may affect the trait. We therefore compared the percentage of puffy fruit between randomly selected F3 families to the percentage of puffy fruit in families selected either by MAS, GS, or the combination of MAS and GS (GS and MAS) under two different selection intensities (Figure 5). Tomato families selected by either MAS, GS, or GS and MAS had a lower percentage of puffy fruit compared to randomly selected families, regardless of the selection intensity (k = 1.66 or k = 0.80) tested (Figure 5). The GS model had high predictability, as demonstrated by cross-validation (r = 0.27, *p* = 7.23 × 10^−4^). The GS model developed from the RIL also exhibited promising results when applied to progeny in the validation population. We detected a significant correlation (accuracy) of r = 0.52 (*p* = 2.36 × 10^−12^) between predicted GEBVs and experimental measurements of puffy fruit in the F3 progeny. However, MAS for the markers solcap_snp_sl_20440 and solcap_snp_sl_18619 was as effective as GS and GS and MAS, with no statistical difference between the strategies. Using GS instead of MAS may prevent the selection of outliers with a high percentage of puffy fruit, as suggested in Figure 5b.

## 4. Discussion

Fruit puffiness has an impact on tomato fruit quality, affecting both processing and fresh market classes. Binary scoring and distribution of data required more than one statistical approach for QTL mapping. The residuals for the percentage of puffy fruit were not normally distributed in this study. We therefore mapped QTLs for the incidence of puffy fruit through interval mapping using a binary method (IM-BI) and the percentage of puffy fruit using non-parametric interval mapping (IM-NP) and parametric composite interval mapping (CIM). Single environment QTL analysis and a joint analysis using data for the two locations were also used to determine whether QTLs for fruit puffiness were detected within and across locations. QTLs were detected on Chr 1 and 2 within the Fremont location and across environments using multiple mapping techniques. Differences in QTL detection were noted for the parametric CIM method, which detected a QTL on Chr 4 and failed to detect the QTL on Chr 1 for data from Fremont. CIM usually eliminates ghost QTLs [36] and has more mapping power and precision than standard interval mapping [37]. However, CIM should be used and interpreted with caution when departure from data normality is substantial, as in our dataset [38]. QTLs for fruit puffiness in the RIL population were not detected in Wooster. Both incidence and severity were higher in Fremont, suggesting that selection against puffy lines based on phenotype will require an environment that favors trait expression. These analyses provide evidence for putative QTLs for fruit puffiness on tomato chromosomes 1, 2, and 4 and a potential interaction increasing fruit puffiness up to 15% between the loci on Chr 2 and 4.

Validation efforts using F3 families derived from further crossing of RILs and parent material confirmed only the QTL on Chr 1, which explained 22.5% of the total variation for the percentage of puffy fruit. QTLs for fruit puffiness were previously reported [6,39,40,41,42,43], but none of these were located on Chr 1. Differences in the QTL number and location are expected when fruit puffiness alleles come from different genetic backgrounds. Fruit puffiness alleles in previous studies were donated by wild tomato parents, while FG02-188, a cultivated elite line, was the source of fruit puffiness alleles described here. In addition, previously described QTLs explained a small proportion of the variation, suggesting that fruit puffiness might be a polygenic trait with many loci of small effect and highly influenced by environmental conditions.

The reproducibility of QTL on Chr 1 across locations, years, and generations suggests the possible use of MAS to create puffiness-resistant lines in similar pedigrees. Due to the low PVE and QTL levels that were detected in some of the analyses, we also considered a GS selection strategy. GS is promoted as better suited for complex traits because it uses genome-wide marker coverage to capture the effect of many small-effect trait-related genes across the genome in an attempt to explain the total genetic variation [44]. GS for decreased fruit puffiness exhibited promising results with the training model developed from the RIL applied to progeny in the validation population. However, the percentage of puffy fruit for validation lines selected through either MAS or GS was statistically the same, and combining GS and MAS did not improve selection over using MAS and GS strategies alone. All 16 or 32 lines (k = 1.66 or k = 0.80) selected through GS or GS and MAS belonged to the sub-population with the lowest percentage of puffy fruit (2K20-8357), for which all individuals were homozygous for the allele B of solcap_snp_sl_20440 and solcap_snp_sl_18619 markers on chromosome 1. These are the same lines that would be selected through MAS alone, and markers associated with the QTL on Chr 1 had large effects in the GS model and influenced GEBV predictions. MAS for decreased fruit puffiness on Chr 1 must therefore seek to eliminate FG02-188 alleles.

## 5. Conclusions

Alleles identified in an elite tomato line were found to be the cause of elevated fruit puffiness in advanced tomato breeding populations. The evidence pointed to the discovery of putative QTLs on chromosomes 1, 2, and 4 and a potential interaction between loci on chromosomes 2 and 4, increasing the percentage of puffy fruit up to 15%. From these putative QTLs, only that on Chr 1 was reproducible across populations and mapping approaches. QTL on Chr 1 is dominant towards increased fruit puffiness and accounts for 5 to 22.5% of the total variation in the percentage of puffy fruit. Approaches using either MAS or GS are effective for selecting against puffy lines compared to when no selection strategy is used at all. Missing heritability issues and the effectiveness of MAS suggest that fruit puffiness in our breeding populations is controlled by a major QTL on Chr 1 and several other small-effect QTLs located throughout the genome.

## Figures and Tables

**Figure 1 plants-13-01454-f001:**
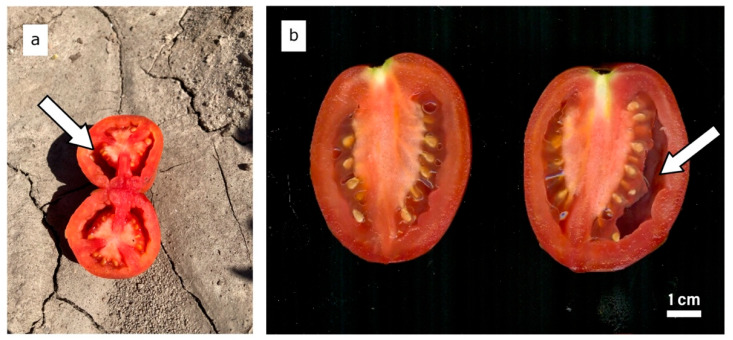
Cross-section of a puffy tomato fruit in the field (**a**). Longitudinal section of a normal tomato fruit on the left and a puffy tomato fruit on the right (**b**). Arrows point to air-filled “puffy” locule.

**Figure 2 plants-13-01454-f002:**
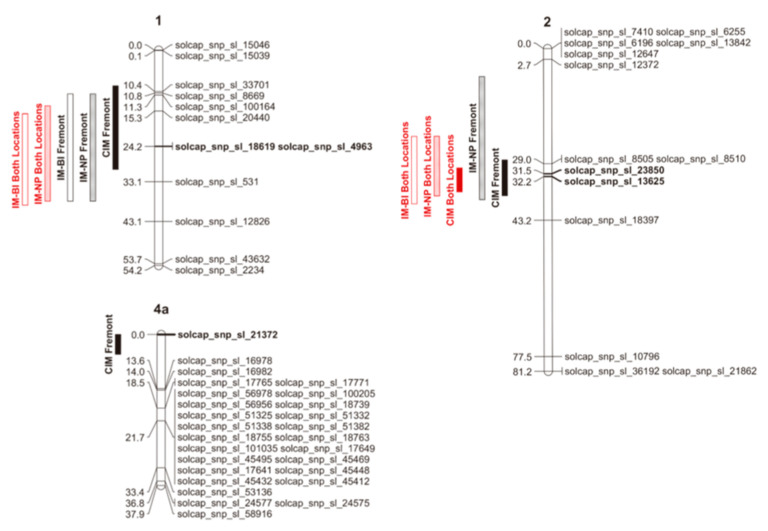
Schematic representations of tomato chromosomes 1, 2, and 4a based on genetic linkage map results, showing 1-LOD support intervals (LSIs) for all detected QTLs using different mapping approaches: IM-BI = interval mapping for the incidence of puffy fruit using a binary model; IM-NP = interval mapping for the percentage of puffy fruit using a non-parametric model; CIM = composite interval mapping for the percentage of puffy fruit using Haley–Knott regression as the solution-generating algorithm. Markers in bold appear within 1-LSI QTL intervals of all mapping approaches. Chromosome representations were created using MapChart software version 2.32 [35].

**Figure 3 plants-13-01454-f003:**
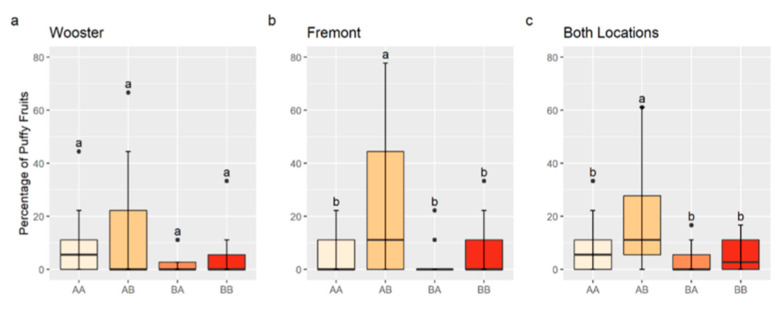
Boxplots for the percentage of puffy fruit by chromosome 2 and 4 genotypes for Wooster (**a**), Fremont (**b**), and data for both locations combined (**c**). AA indicates a homozygote for the allele A of solcap_snp_sl_13625 (chromosome 2) and solcap_snp_sl_21372 (chromosome 4); AB indicates a homozygote for the allele A of solcap_snp_sl_13625 and a homozygote for the allele B of solcap_snp_sl_21372; BA indicates a homozygote for the allele B of solcap_snp_sl_13625 and a homozygote for the allele A of solcap_snp_sl_21372; and BB indicates a homozygote for the allele B of solcap_snp_sl_13625 and solcap_snp_sl_21372. Different letters indicate statistically significant differences among genotypes by the Tukey’s Honest Significant Difference Test (*p* < 0.05).

**Figure 4 plants-13-01454-f004:**
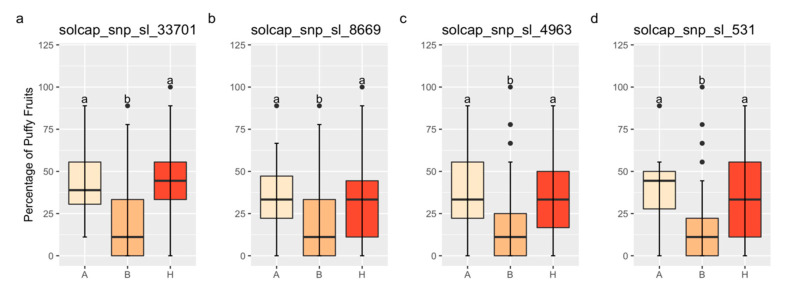
Boxplots for the percentage of puffy fruit by genotype for the validation populations. “H” means heterozygotes, and “A” and “B” indicates homozygotes for the FG02-188 and Ohio 2K9-5533-1 alleles, respectively. The markers solcap_snp_sl_33701 (**a**), solcap_snp_sl_8669 (**b**), solcap_snp_sl_4963 (**c**), and solcap_snp_sl_531 (**d**), located on chromosome 1, are validated. Same lower-case letters mean that genotype classes did not differ by the Tukey’s Honest Significant Difference Test (*p* < 0.05).

**Figure 5 plants-13-01454-f005:**
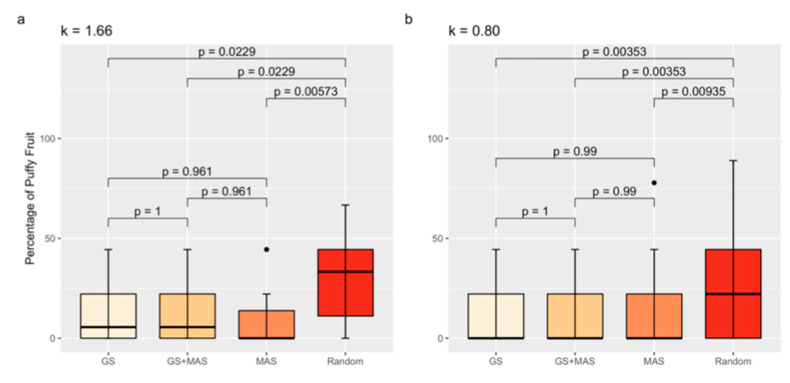
Percentage of puffy fruit averages by selection strategy group for k = 1.66 (**a**) and k = 0.80 (**b**). Boxplots are color coded to distinguish GS = genomic selection; MAS = marker-assisted selection for the markers solcap_snp_sl_20440 and solcap_snp_sl_18619. Black dots are outliers. Tukey’s Honest Significant Difference Test was used to compare means. *p*-values are shown.

**Table 1 plants-13-01454-t001:** Genetic linkage map for the Cmm-RIL population.

					Genetic Map vs. Physical Map (Sl4.0) Correlation		
Chr	Number of Markers	Chr Length (cM)	Average Distance between Markers (cM)	Largest Distance between Markers (cM)	^a^ *p*-Value	^b^ R^2^	^c^ ρ
1	12	54.2	4.9	10.6	0.0001	0.7581	1.0000
2	14	81.2	6.2	34.3	0.0000	0.7522	1.0000
3	6	41	8.2	28.7	0.0006	0.9509	1.0000
4a	27	37.9	1.5	13.6	0.0000	0.4790	1.0000
4b	3	0.5	0.2	0.5	0.2305	0.7491	1.0000
5	16	35	2.3	9.9	0.0000	0.7728	1.0000
6	9	93.3	11.7	46.8	0.0001	0.8960	1.0000
7	11	47.7	4.8	17.2	0.0002	0.7813	1.0000
8	7	55.1	9.2	33	0.0061	0.7671	1.0000
9	13	62.6	5.2	51.4	0.0000	0.9945	1.0000
10	10	63.6	7.1	31	0.0012	0.7198	1.0000
11	3	11.1	5.6	9.1	0.5260	−0.0803	1.0000
12	5	14.9	3.7	8.2	0.1730	0.3516	1.0000
**Overall**	136	598	4.9	51.4			

Chr = chromosome. ^a,b^ *p*-values and Determination coefficients (R^2^) for the linear regression ANOVA and ^c^ Spearman’s rank correlation coefficients between marker position in the genetic linkage map and in the physical map.

**Table 2 plants-13-01454-t002:** QTL mapping results for the Cmm-RIL population.

Location	Variable	Mapping Method	Chr	QTL Peak Position (cM)	LOD Peak	^a^ LOD Cutoff	^b^ Nearest Marker	^c^ PVE	^d^ Additive Effect	^e^ Odds Ratio	^f^ Beneficial Allele	^g^ Physical Position of Nearest Marker (bp)
Fremont	Percentage of puffy fruit	CIM	1	17	2.69	2.46	solcap_snp_sl_20440	6.40	−4.40	-	B	3,637,274
			2	32	3.06	2.46	solcap_snp_sl_13625	8.30	−5.00	-	B	40,480,064
			4	0	2.83	2.46	solcap_snp_sl_21372	7.10	4.70	-	A	2,951,634
	Percentage of puffy fruit	IM-NP	1	20	2.86	2.40	solcap_snp_sl_18619	5.00	−3.90	-	B	68,611,189
			2	32	2.63	2.40	solcap_snp_sl_13625	8.30	−5.00	-	B	40,480,064
	Incidence of puffy fruit	IM-BI	1	21	2.67	2.48	solcap_snp_sl_18619	-	-	3.40	B	68,611,189
Both Locations	Percentage of puffy fruit	CIM	2	32	3.59	2.40	solcap_snp_sl_13625	9.50	−4.10	-	B	40,480,064
	Percentage of puffy fruit	IM-NP	1	29	3.50	2.46	solcap_snp_sl_531	5.40	−3.10	-	B	73,097,261
			2	32	3.18	2.46	solcap_snp_sl_13625	9.50	−4.10	-	B	40,480,064
	Incidence of puffy fruit	IM-BI	1	28	3.25	2.49	solcap_snp_sl_4963	-	-	3.60	B	70,083,502
			2	32	2.55	2.49	solcap_snp_sl_13625	-	-	3.90	B	40,480,064

CIM = composite interval mapping; IM-NP = interval mapping using a non-parametric model; IM-BI = interval mapping using a binary model; Chr = chromosome. ^a^ LOD cutoffs were defined through permutation tests (α = 0.05, n = 1000; Churchill and Doerge, 1994). ^b^ Nearest marker refers to the nearest marker to the QTL peak position (cM). ^c^ PVE is the percentage of variance explained by a QTL based on the nearest marker to the QTL peak position (cM) estimated by the formula 1–10^−2 LOD/n^, where n is the sample size and LOD is the LOD score. ^d^ The additive effect corresponds to half the difference between the phenotype averages for the two homozygotes. ^e^ Odds ratios estimated by $e^{\beta_{1}}$, where e is the natural base and equals to 2.718, and $\beta_{1}$ is the regression coefficient of the logistic regression for the homozygote A. The odds ratio for the incidence of puffy fruit in Fremont means that the odds of having puffy fruit for homozygotes A on chromosome 1 are 3.4 to 3.6 times the odds for homozygotes B. ^f^ The allele responsible for decreased fruit puffiness in the RIL population in which “A” is the allele donated by the parent FG02-188 and “B” is the allele donated by the parent Ohio 2K9-5533-1. ^g^ Physical position of nearest marker (bp) in the Tomato Genome version SL4.0 [25].

**Table 3 plants-13-01454-t003:** *p*-values of ANOVA’s F-tests from marker–trait regressions for the validation populations studied separately and combined. Selected markers are those identified by interval mapping as potentially linked to QTLs for percentage of puffy fruit in the mapping population.

		*p*-Values of ANOVA F Tests from Marker–Trait Regressions			
Chr	Marker	Validation Population	2K20-8312 Sub-Population	2K20-8322 Sub-Population	2K20-8357 Sub-Population
1	solcap_snp_sl_33701	5.59 × 10^−7^	0.83	monomorphic B	monomorphic B
1	solcap_snp_sl_8669	1.04 × 10^−4^	0.71	0.33	monomorphic B
1	solcap_snp_sl_100164	1.32 × 10^−9^	monomorphic A	0.31	monomorphic B
1	solcap_snp_sl_20440	3.02 × 10^−11^	monomorphic A	monomorphic A	monomorphic B
1	solcap_snp_sl_18619	3.02 × 10^−11^	monomorphic A	monomorphic A	monomorphic B
1	solcap_snp_sl_4963	4.54 × 10^−7^	0.85	0.45	monomorphic B
1	solcap_snp_sl_531	5.05 × 10^−9^	0.75	0.04	monomorphic B
2	solcap_snp_sl_8505	0.80	monomorphic A	0.92	monomorphic A
2	solcap_snp_sl_8510	0.80	monomorphic A	0.92	monomorphic A
2	solcap_snp_sl_23850	0.68	monomorphic A	0.25	monomorphic A
2	solcap_snp_sl_13625	0.76	monomorphic A	0.59	monomorphic A
4	solcap_snp_sl_21372	0.94	0.44	0.94	0.36

Monomorphic A and B = monomorphic for the allele “A” donated by the parent FG02-188 and for the allele “B” donated by the parent Ohio 2K9-5533-1.

## Data Availability

The data that support the findings of this study are available on request from the corresponding author.

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
