# Peer review of "Identification and Validation of Quantitative Trait Loci Associated with Fruit Puffiness in a Processing Tomato Population"

_plants, 2024, doi:10.3390/plants13111454_

Round 1
Reviewer 1 Report
Comments and Suggestions for Authors
This study discusses the impact of physiological disorders on tomato yield and quality, focusing on fruit puffiness caused by cavities inside the locule. The manuscript identifies QTLs for fruit puffiness on Chromosomes 1, 2, and 4, with the QTL on Chromosome 1 explaining up to 22.5% of the variance in puffy fruit. They discuss that Marker-assisted selection (MAS) for the QTL on Chromosome 1 is as efficient as genomic selection (GS) in reducing the incidence and severity of puffy fruit.
Overall, the research and experiments are clearly described. The analysis of the published data was provided with a sufficient level of scientific novelty. But, there are important flaws in the manuscript listed below:
-The manuscript has moderate grammar and punctuation problems and it needs to be checked for the English language by a native speaker. In the abstract, line 11, “we used a recombinant inbred line (RIL)” could be replaced with “we used recombinant inbred lines (RILs)”. In the last paragraph of the introduction, line 69, the sentence “QTL interactions increasing fruit puffiness severity were detected.” could be replaced with "QTL interactions that increase fruit puffiness severity were detected”.
-In the last part of the introduction, lines 66-72, the goals of the research need to be mentioned. Please rewrite this paragraph.
- In section 2.1. mapping population, more explanation about the populations and the reasons for using these populations are needed.
- In Fig. 5 I recommend using the mean compassion letter (such as Tukey’s Honest Significant Difference Test) instead of p-value at the top of the figure.
-All scientific names of species in references need to be italicized.
Comments on the Quality of English Language
The manuscript has moderate grammar and punctuation problems and it needs to be checked for the English language by a native speaker.
Author Response
Comments and Suggestions for Authors
This study discusses the impact of physiological disorders on tomato yield and quality, focusing on fruit puffiness caused by cavities inside the locule. The manuscript identifies QTLs for fruit puffiness on Chromosomes 1, 2, and 4, with the QTL on Chromosome 1 explaining up to 22.5% of the variance in puffy fruit. They discuss that Marker-assisted selection (MAS) for the QTL on Chromosome 1 is as efficient as genomic selection (GS) in reducing the incidence and severity of puffy fruit.
Overall, the research and experiments are clearly described. The analysis of the published data was provided with a sufficient level of scientific novelty. But, there are important flaws in the manuscript listed below:
-The manuscript has moderate grammar and punctuation problems and it needs to be checked for the English language by a native speaker. In the abstract, line 11, “we used a recombinant inbred line (RIL)” could be replaced with “we used recombinant inbred lines (RILs)”. In the last paragraph of the introduction, line 69, the sentence “QTL interactions increasing fruit puffiness severity were detected.” could be replaced with "QTL interactions that increase fruit puffiness severity were detected”.
Response: The corresponding author is a native speaker and he has revised the manuscript. Specifically, we have rewritten and refined the abstract, rewritten the last paragraph of the introduction (see comment, below), revised figure 1, and modified the discussion.
-In the last part of the introduction, lines 66-72, the goals of the research need to be mentioned. Please rewrite this paragraph.
Response: Done. We added the statement “Our specific goals were to describe the genetics underlying an important fruit quality trait and compare selection strategies” to the beginning of the final paragraph of the introduction.
In section 2.1. mapping population, more explanation about the populations and the reasons for using these populations are needed.
Response: It was unclear to us what further explanation the reviewer wanted. We added the statement “for evaluation and validation” to clarify the purpose of crosses derived from the initial mapping population.
In Fig. 5 I recommend using the mean compassion letter (such as Tukey’s Honest Significant Difference Test) instead of p-value at the top of the figure.
Response: We feel that having the p-value is more informative than a mean separation letter because the value communicates the level of significance. We prefer to leave this notation.
-All scientific names of species in references need to be italicized.
Response: Reviewed and corrected.
Comments on the Quality of English Language
The manuscript has moderate grammar and punctuation problems and it needs to be checked for the English language by a native speaker.
Response: The corresponding author is a native speaker.
Reviewer 2 Report
Comments and Suggestions for Authors
Dear authors and the editor,
Major concerns:
1. The study highlights the dominance of the allele from the elite tomato line FG02-188 in increasing fruit puffiness, providing valuable insights into the genetic architecture of this trait.
2. The validation of the QTL in subsequent generations demonstrates the reproducibility of the findings.
3. The paper explores the application of marker-assisted selection and genomic selection for reducing fruit puffiness, offering promising breeding strategies.
Minor concerns and comments:
Title: The title should be more specific, and better pointed out the key QTL, such as qFP1 (the QTL on Chr 1 ), a potential locus for regulating the Fruit Puffiness in tomato.
Abstract: It is too long and detailed. It should be concise, summarizing the main findings and their significance. This part lacks of highlights and new findings.
some questions to be answered:
1. Limited Heritability: The low heritability of fruit puffiness suggests the existence of additional minor QTLs or environmental factors that were not fully captured in this study.
2. QTL Validation: The failure to validate the QTLs on chromosomes 2 and 4 may warrant further exploration into the potential interaction effects.
3. Environmental Influence: The influence of environmental factors on fruit puffiness was not thoroughly investigated, necessitating further research.
Specific suggestions for improvement:
1. Abstract Refinement: Refine the abstract to more clearly emphasize the main research question, findings, and significance.
2. Introduction Restructuring: Restructure the introduction to more directly lead into the specific research question addressed in this study.
3. Methods Expansion: Expand the methods section to include more detail on experimental design, phenotyping, and genotyping protocols.
4. Figure Enhancements: Enhance figure 1 to better illustrate fruit puffiness and consider adding a figure showing the genetic map.
5. Results Clarification: Clarify the results by specifying if the QTLs are unique or if they overlap with previously reported QTLs for other traits.
6. Discussion Expansion: Expand the discussion to consider potential reasons for the low heritability and the failure to validate some QTLs.
Overall, this paper makes a valuable contribution to understanding the genetics of fruit puffiness in tomatoes. However, there are opportunities to enhance clarity, detail, and focus, particularly in the abstract, introduction, methods, and results sections. Additionally, exploring the broader genetic architecture and environmental influences would further strengthen the study.
Comments on the Quality of English LanguageDear authors and the editor,
Major concerns:
1. The study highlights the dominance of the allele from the elite tomato line FG02-188 in increasing fruit puffiness, providing valuable insights into the genetic architecture of this trait.
2. The validation of the QTL in subsequent generations demonstrates the reproducibility of the findings.
3. The paper explores the application of marker-assisted selection and genomic selection for reducing fruit puffiness, offering promising breeding strategies.
Minor concerns and comments:
Title: The title should be more specific, and better pointed out the key QTL, such as qFP1 (the QTL on Chr 1 ), a potential locus for regulating the Fruit Puffiness in tomato.
Abstract: It is too long and detailed. It should be concise, summarizing the main findings and their significance. This part lacks of highlights and new findings.
some questions to be answered:
1. Limited Heritability: The low heritability of fruit puffiness suggests the existence of additional minor QTLs or environmental factors that were not fully captured in this study.
2. QTL Validation: The failure to validate the QTLs on chromosomes 2 and 4 may warrant further exploration into the potential interaction effects.
3. Environmental Influence: The influence of environmental factors on fruit puffiness was not thoroughly investigated, necessitating further research.
Specific suggestions for improvement:
1. Abstract Refinement: Refine the abstract to more clearly emphasize the main research question, findings, and significance.
2. Introduction Restructuring: Restructure the introduction to more directly lead into the specific research question addressed in this study.
3. Methods Expansion: Expand the methods section to include more detail on experimental design, phenotyping, and genotyping protocols.
4. Figure Enhancements: Enhance figure 1 to better illustrate fruit puffiness and consider adding a figure showing the genetic map.
5. Results Clarification: Clarify the results by specifying if the QTLs are unique or if they overlap with previously reported QTLs for other traits.
6. Discussion Expansion: Expand the discussion to consider potential reasons for the low heritability and the failure to validate some QTLs.
Overall, this paper makes a valuable contribution to understanding the genetics of fruit puffiness in tomatoes. However, there are opportunities to enhance clarity, detail, and focus, particularly in the abstract, introduction, methods, and results sections. Additionally, exploring the broader genetic architecture and environmental influences would further strengthen the study.
Author Response
Comments and Suggestions for Authors
Dear authors and the editor,
Major concerns:
Response: As stated, these are not concerns but positive comments.
1. The study highlights the dominance of the allele from the elite tomato line FG02-188 in increasing fruit puffiness, providing valuable insights into the genetic architecture of this trait.
2. The validation of the QTL in subsequent generations demonstrates the reproducibility of the findings.
3. The paper explores the application of marker-assisted selection and genomic selection for reducing fruit puffiness, offering promising breeding strategies.
Response: We feel that these comments are important and supports the rigor of our approach as providing insight, reproducible information, and promising strategies.
Minor concerns and comments:
Title: The title should be more specific, and better pointed out the key QTL, such as qFP1 (the QTL on Chr 1 ), a potential locus for regulating the Fruit Puffiness in tomato.
Abstract: It is too long and detailed. It should be concise, summarizing the main findings and their significance. This part lacks of highlights and new findings.
Response: We have rewritten and refined the abstract.
some questions to be answered:
Limited Heritability: The low heritability of fruit puffiness suggests the existence of additional minor QTLs or environmental factors that were not fully captured in this study.
Response: This point is exactly why we compared MAS to Genomic Selection. We agree that there may be other minor (sub-significant threshold) QTL contributing to Puffy fruit, but we also feel that we have captured a major locus and have demonstrated the effectiveness of selection.
2. QTL Validation: The failure to validate the QTLs on chromosomes 2 and 4 may warrant further exploration into the potential interaction effects.
Response: QTL mapping is subject to both environmental variation and false positives. In addition our validation populations also introduced new genetic backgrounds which may have affected our ability to reproducibly detect the QTL on chromosomes 2 and 4. We have added this explanation to the text.
A key point that I would like the reviewers and editors to keep in mind is that we created subsequent generations, evaluated these, and demonstrated that the QTL on chromosome 1 was robust. Further, because of the possibility that other QTL (chromosome 2 and 4, and possibly others) were falling below significant thresholds we compared MAS to GS approaches.
3. Environmental Influence: The influence of environmental factors on fruit puffiness was not thoroughly investigated, necessitating further research.
Response: Understanding the role of environmental influence was not the goal of the study.
Specific suggestions for improvement:
Response: These are helpful suggestions. Specific responses follow.
Abstract Refinement: Refine the abstract to more clearly emphasize the main research question, findings, and significance. Response: We have rewritten and refined the abstract.
2. Introduction Restructuring: Restructure the introduction to more directly lead into the specific research question addressed in this study.
Response: We have addressed this suggestion primarily by revising the final paragraph to be more direct in the goals and outcomes.
3. Methods Expansion: Expand the methods section to include more detail on experimental design, phenotyping, and genotyping protocols.
Response: It is rare to be asked to expand a MS. To address this comment we rearranged the methods to keep phenotypic data collection and analysis together while also keeping the genetic map and QTL analysis together.
4. Figure Enhancements: Enhance figure 1 to better illustrate fruit puffiness and consider adding a figure showing the genetic map.
Response: We explored a full genetic map of all 12 chromosomes, but believe that highlighting only chromosomes with significant QTL serves to highlight the reproducibility and map coverage in those regions. The remaining of the genetic map is effectively summarized in table 1, with information relevant to quality of coverage and co-linearity with the physical map.
5. Results Clarification: Clarify the results by specifying if the QTLs are unique or if they overlap with previously reported QTLs for other traits.
Response: To our knowledge these are unique QTL for fruit puffiness. We discuss QTL from wild species which contribute to puffiness.
6. Discussion Expansion: Expand the discussion to consider potential reasons for the low heritability and the failure to validate some QTLs.
Response: We edited the discussion to emphasize the role of statistical methods, location differences, and genetic background differences in detecting a QTL. A key conclusion is that the Chr. 1 QTL is reproducible and validated across generations. Selection against the FG02-188 allele is effective.
Overall, this paper makes a valuable contribution to understanding the genetics of fruit puffiness in tomatoes. However, there are opportunities to enhance clarity, detail, and focus, particularly in the abstract, introduction, methods, and results sections. Additionally, exploring the broader genetic architecture and environmental influences would further strengthen the study.
Response: We appreciate this comment.